# Tribology of EDM Recast Layers Vis-À-Vis TIG Cladding Coatings: An Experimental Investigation

**DOI:** 10.3390/mi16080913

**Published:** 2025-08-07

**Authors:** Muhammad Adnan, Waqar Qureshi, Muhammad Umer

**Affiliations:** 1Mechanical Engineering Department, University of Engineering and Technology Taxila, Taxila 47050, Pakistan; waqar.ahmed@uettaxila.edu.pk; 2Department of Mechanical, Aeronautical and Biomedical Engineering and the Materials and Surface Science Institute, University of Limerick, V94 T9PX Limerick, Ireland; muhammad.umer2@ul.ie

**Keywords:** tribological performance, EDM, coatings, difficult-to-cut material, taguchi analysis, parameter optimization, microhardness, wear resistance, wear mechanism

## Abstract

Tribological performance is critical for the longevity and efficiency of machined components in industries such as aerospace, automotive, and biomedical. This study investigates whether electrical discharge machining recast layers can serve as a cost-effective and time-efficient alternative to conventional tungsten inert gas cladding coatings for enhancing surface properties. The samples were prepared using electrical discharge machining and tungsten inert gas cladding. For electrical discharge machining, various combinations of electrical and non-electrical parameters were applied using Taguchi’s L18 orthogonal array. Similarly, tungsten inert gas cladding coatings were prepared using a suitable combination of current, voltage, powder size, and speed. The samples were characterized using, scanning electron microscopy, optical microscopy, microhardness testing, tribological testing, energy-dispersive X-ray spectroscopy, X-ray diffraction analysis and profilometry. The electrical discharge machining recast layers exhibited superior tribological performance compared to tungsten inert gas cladding coatings. This improvement is attributed to the formation of carbides, such as TiC and Ti_6_C_3.75_. The coefficient of friction and specific wear rate were reduced by 11.11% and 1.57%, respectively, while microhardness increased by 10.93%. Abrasive wear was identified as the predominant wear mechanism. This study systematically compares electrical discharge machining recast layers with tungsten inert gas cladding coatings. The findings suggest that optimized electrical discharge machining recast layers can serve as effective coatings, offering cost and time savings.

## 1. Introduction

Surface coatings play a vital role in enhancing the tribological performance of machined components, which are critical in industries such as aerospace, automotive, and biomedical. While tungsten inert gas (TIG) cladding has been widely used for this purpose, it is often associated with high costs and time-consuming processes. This study explores whether electrical discharge machining (EDM) recast layers can serve as a more efficient alternative, offering comparable or superior tribological performance while reducing costs and processing time.

Tribological performance—encompassing friction, wear, and lubrication—is crucial for components that undergo surface-to-surface contact. Microhardness, a measure of a material’s resistance to deformation, and specific wear rate, which quantifies material loss during wear, are key indicators of a coating’s effectiveness. Improved tribological performance leads to reliable, safe, and energy-efficient mechanical systems [1,2]. Various coating techniques, including TIG cladding, are employed to enhance the tribological performance of machined components.

TIG cladding is a composite coating applied through a TIG torch surface melting technique. It is used in industries such as automotive, biomedical, and aerospace. In these industries, components frequently experience surface contact, leading to excessive heat and friction, which can cause rapid damage [3]. Currently, TIG cladding coatings are used to address these issues. For example, Kumar et al. deposited a hard solid-lubricating coating on a Ti-6Al-4V substrate using pre-placed powder, enhancing tribological performance, mechanical properties, and corrosion resistance [4]. An et al. obtained a high-quality TIG cladding coating with superior microhardness and concluded that low-cost coatings can be achieved using pre-sintered composites [5]. Other researchers have found that TIG current significantly affects the hardness and tribological performance of coatings [6,7]. Debta et al. studied how stand-off distance affects the tribological performance and microstructure of TIG cladding coatings [8]. Maleque et al. reported that TIG cladding significantly improves the hardness and wear resistance of substrate surfaces, with voltage and powder size being key parameters [3]. However, TIG cladding has limitations, including high capital costs, specialized surface preparation, significant time investment, and advanced equipment. Thus, this study aims to reduce the costs and time required for post-machining surface coatings by exploring the potential of EDM recast layers as an alternative.

EDM is widely used for processing conductive materials. It has applications in aerospace, cutting tools, automotive, bearings, surgical instruments, and mold industries. Also, low machinable materials and geometrically intricate machine parts can be easily processed by EDM [9,10]. Moreover, due to the difficult-to-cut nature of substrate material, EDM is preferred for its machining [11].

Recently, researchers have explored EDM potential for improving tribological performance. For instance, Manderna et al. improved the wear and corrosion behavior of surfaces using wire EDM, though they did not compare these improvements with TIG cladding coatings [12]. Joshy et al. added tungsten disulphide powder to the dielectric medium in a powder-mixed EDM process. This change improved the coefficient of friction (COF) and specific wear rate [13]. Elaiyarasan et al. applied composite electrode-assisted electrical discharge coating to enhance tribological performance, studying the effects of normal force, sliding speed, and sliding time on COF and wear rate [14]. Similarly, micro-EDM has been used to improve the tribological performance of miniaturized components [15]. Other researchers have observed improvements in tribological behavior through die-sinking EDM [16,17]. However, these studies primarily compare the tribological performance of EDM-treated surfaces with the substrate material, rather than conventional coatings like TIG cladding.

To the best of the authors’ knowledge, no research has systematically compared the tribological performance of EDM recast layers—considering both electrical and non-electrical parameter variations—with TIG cladding coatings. Due to the limited information available to develop a framework for using EDM recast layers as effective coatings, a more in-depth study is warranted. Therefore, this study conducts extensive experiments using profilometry, optical microscopy, scanning electron microscopy (SEM), microhardness testing, tribological testing, energy-dispersive X-ray spectroscopy (EDX), and X-ray diffraction analysis (XRD) analysis to address this gap. The findings provide a basis for using EDM recast layers as a viable alternative to conventional coatings, offering cost and time savings.

## 2. Materials and Methods

### 2.1. Materials

Ti-6Al-4V (TC4) was used as the substrate material. It was purchased from Shaanxi Yunzhong Metal Technology Company Limited, Baoji City, China. Its chemical composition was determined using a handheld XRF analyzer (Oxford X-MET 5000, High Wycombe, UK) and optical emission spectroscopy (Spectrotest, Kleve, Germany). The microhardness (H_m_) was measured using a micro-vicker testing machine (810-401D, Mitutoyo, Germany), and surface roughness (R_a_) was measured using a portable surface roughness tester (SJ-410, Mitutoyo, Industry, CA, USA). The observations are presented in Table 1. During EDM, copper and graphite were used as electrode materials, and 406 EDM oil was utilized as the dielectric medium. For TIG cladding coatings, silicon carbide powder, polyvinyl acetate, distilled water, alcohol, pure argon gas, and a tungsten thoriated electrode were used. A diamond pin was chosen as the counter-body for tribological testing. The chemical compositions and specifications of the materials are provided in Table 1.

### 2.2. Experimental Methods

The samples were prepared using EDM and TIG cladding. Then, their characterization was carried out. Each step of the experimentation is discussed separately and shown in Figure 1.

#### 2.2.1. EDM

Before EDM, a lathe machine (Colchester Student 1800, Colchester Machine Tool Solutions, Birmingham, UK) was used to obtain the desired specimen size. Parting operation was done to obtain the length of 25 mm of each specimen. After facing operation, the surface of each specimen was polished with sandpaper up to an average surface roughness of 1 ± 0.01 µm. The recast layer was produced after the EDM process. The design of experiments (DOE) was based on Taguchi’s L18 orthogonal array, incorporating a mixed-level design (3^2^ × 2^1^). In this DOE, two factors—discharge current (I) and pulse on time (Ton)—had three levels, and one factor—electrode material (E)—had two levels. Discharge current and pulse on time were selected as process variables because they directly influence the energy input during EDM, affecting the formation and properties of the recast layer [18]. Copper (Cu) and graphite (Gr) electrodes were chosen as they are frequently used in the mold industry [19]. The machining parameters, based on pilot trials, are illustrated in Table 2.

During EDM, the specimens were clamped in a fixture, and die-sinking EDM (Neuar E30, Taipei, Taiwan) was used for the experimental trials, and its schematic is shown in Figure 2a. Each specimen was machined for 5 min, during which the surface integrity was significantly altered, affecting tribological performance. After each trial, the electrodes were ground with sandpaper to ensure consistent surface roughness and flatness. Each trial was carried out using the same equipment and operator to ensure consistency. Every experiment was repeated twice for precision.

#### 2.2.2. TIG Cladding Coating

For TIG cladding, the specimens were cleaned with acetone to remove contaminants. A mixture of 4 mg silicon carbide powder, 1 mL polyvinyl acetate, distilled water, and alcohol were agitated to form a paste. The paste was applied to the clean surfaces of the specimens and dried to remove moisture using a standard delivery universal oven (UF30, Memmert, Schwabach, Germany) at 60 °C for 30 min. The paste on the substrate surface was melted using a TIG arc cladding machine (Technology TIG 182, Villaverla, Italy). Its schematic has been shown in Figure 2b. A tungsten thoriated electrode was used to produce an arc, and pure argon gas was used for shielding. Input parameters were chosen after extensive experimentation. Cladding current is a significant input parameter. Its higher values favor the hardness and wear resistance of the coating [20]. On the other hand, too high values of the current produce more defects [21]. A suitable value was chosen to make sure that the coating does not burn off. The input variables based on pilot trials have been given in Table 3. The samples were prepared in doublets, and the same operator used the same equipment for all samples to avoid variation in results.

#### 2.2.3. Characterization of Coatings

The surface morphology was inspected using scanning electron microscope (EVO 15, Zeiss, Cambridge, UK). The specimens were thoroughly cleaned to depict the clear images. These were placed on the turntable of SEM. Surface defects were examined during inspection by taking images at 5000× to 30,000× magnifications.

For assessment of recast layer thickness (h_c_) and microstructure, specimens were cross-sectioned perpendicular to the EDM-ed surface using a metallurgical cut-off saw. The mount was made by embedding specimens in cold resin epoxy. Then, grinding, polishing and etching were done. Optical microscope (Leica, Wetszler, Germany) and scanning electron microscope were used. Three to five different regions based on variations in recast layer thickness were picked for each specimen to determine average recast layer thickness. For optimization of recast layer thickness, Taguchi’s optimization criterion with larger is better signal-to-noise ratio was used.

The microhardness of the specimens was determined using a diamond pyramid indenter. A 4.9 N test force-load and 10 s dwell time were applied, following ASTM E384 standard [22]. Three readings at discrete points were taken to calculate the average microhardness. The optimal microhardness was determined using Taguchi’s optimization criterion. Larger is better signal-to-noise ratio was selected for this purpose. Two samples with comparable microhardness (trial 9 and trial 18) were selected for further analysis [23] and compared with TIG cladding coatings and the substrate material.

Tribological testing was performed using a pin-on-disk tester (Microtest, Madrid, Spain), whose schematic and experimental set up have been shown in Figure 3a and Figure 3b, respectively. Four types of samples were tested. Two samples of comparable microhardness were selected from EDM recast layers. A third sample from TIG cladding coating was used. Then, substrate material was used as a fourth sample. Wear behavior of EDM-ed samples was compared to determine the optimized sample. This optimized sample was then compared with the TIG-cladded sample. Also, the comparison of all coated samples was made with the substrate material. In the test configuration, the specimens rotated while sliding against a stationary diamond pin mounted on a 90° cone. Testing was conducted at room temperature under unlubricated conditions. Normally, third body wear also affects the results [24]. But, in the given test configuration, the primary wear mode was abrasive, and sufficient space was available for particles ejection. Hence, no third body effects were considered. Usually, tribological tests are carried out for a specific application. EDM and TIG cladding have versatile applications such as automotive, biomedical and aerospace industries. In this regard, many combinations of counter-body material, applied load, track radius, rotational speed, and sliding distance were possible. As the indentation exhibits the wear response of the recast layers rather than specific application. Several experiments were performed to choose the experimental parameters. It was intended to obtain a measurable and suitable wear section from the heat-affected zone. The experimental conditions based on pilot trials have been provided in Table 4. Before testing, sensor calibration was performed, and the wear volume and COF were measured. Each test was repeated thrice for fair assessment, and wear testing was performed twice on doublet samples. After tribological testing, worn surface analysis was conducted using SEM equipped with SmartEDX (EVO 15, Zeiss, Cambridge, UK). High-magnification images were taken to identify wear mechanisms and sub-mechanisms.

The chemical composition of worn products was determined using SmartEDX, with four different areas analyzed to calculate the average chemical composition. X-ray diffraction analysis was conducted using an X-ray diffractometer (Proto, Waterloo, ON, Canada). Prior to analysis, the diffractometer was calibrated, and the diffraction angle was varied from 20° to 90° to distinguish different phases of worn products. The graphs were drawn using Origin software. Then, JCPD cards were checked from MDI Jade 6 and labeled on graphs.

Finally, surface roughness of the coated surfaces was measured using a portable surface roughness tester. Three readings were noted down at three distinct points to calculate average surface roughness. Taguchi’s optimization criterion was used to determine the optimal surface roughness, and the statistical software Minitab 22 was employed to study the effect of input parameters on the output response. For optimization purposes, the selected signal-to-noise ratio was smaller is better.

## 3. Results and Discussion

### 3.1. Surface Morphology

SEM images were taken at higher magnification for the surface morphology of the coated specimens and substrate material, as presented in Figure 4. Surface pits and globules were observed on the surface of the copper-coated specimen. Less surface cracks were also seen. For graphite-coated sample, more cracks were found on the coated surface. This verifies that surface cracking directly depends upon the amount of carbon contents present on the coated surface. Blackish contents were seen in the cavities. These were because of the decomposition of graphite and the cracking of dielectric, as found by [25]. Micro-porosity and some pock marks were also observed on the coated surface. The pits, globules, micro-cracks, and cavities were produced due to the repetitive melting cycles, cooling cycles, and low dielectric flushing efficiency during EDM. The melted material re-solidified and became firm on the surface of the specimen, resulting in the above-mentioned surface defects [26]. The TIG-cladded sample was more uniform and compact as compared to the EDM-ed samples. It exhibited less porosity and pitting. On the other hand, the substrate material had no surface defects such as pits, cavities, globules, etc. The difference in the surface morphology of EDM-ed specimens and the TIG-cladded sample confirmed the viability of EDM recast layers as effective coatings.

### 3.2. Recast Layer Thickness and Microstructure

To determine the recast layer thickness and microstructure, EDM-ed samples and TIG-cladded sample were cross-sectioned as shown in Figure 5a–c. After EDM, three different layers are formed on the surface being machined (Figure 5d). A heavily alloyed, non-etchable recast layer is formed due to the rapid solidification of melted material. It is also known as a white layer and has a different metallographic and microstructure than the substrate material. The frequent cyclic heating and quenching routines of the process form a heat-affected zone (HAZ) below the recast layer. This is a slightly tempered layer. The heat from the HAZ retreats slowly as compared to the white layer. Below the HAZ, there is bulk material [27]. Out of these affected layers, the recast layer can produce unique characteristics to the substrate material. Its thickness depends upon variation in EDM parameters. It increases with an increase in pulse on time because of more heat transfer into the specimen material. Also, the flushing pressure remains constant, and the dielectric cannot clear away the molten material. This melted material penetrates into the interior of the specimen, resulting in a thick recast layer. The recast layer thickness also increases with an increase in discharge current as the substrate material reaches its melting point quickly. The vaporized and melted material forms a thick recast layer [19,28]. The graphite electrode produced thicker recast layers than copper electrode, especially at higher values of discharge current. This was due to the porous nature of graphite. Another possible reason was more cracking at high discharge energy. These observations have been shown in Figure 6a. The EDM-ed surfaces produced a recast layer over the substrate with a sandwiched HAZ. The thickness of the HAZ depends on how heat is transferred between the workpiece and the dielectric fluid. A forced flushing through the external nozzle was enabled to enhance the dielectric flushing efficiency. So, the maximum thickness of HAZ was limited to 55 μm only.

Taguchi optimization of recast layer thickness was performed. Pulse on time was ranked one with 53.17% contribution. Discharge current and electrode material were ranked second and third with 29.51% and 17.31% contributions, respectively. The optimal recast layer thickness was produced from graphite electrode at 20 A discharge current and 400 μs pulse on time (see Figure 6b and Table 5). On the other hand, the average recast layer thickness of the TIG-cladded sample was 194.86 ± 5 µm. The optimized EDM-ed sample, even with 10.19% less recast layer thickness, performed tribologically better than the TIG-cladded sample (Figure 6c).

As far as the microstructure is concerned, EDM-ed samples had hexagonal martensitic α’ microstructures. This α′ microstructure was composed of long orthogonally oriented martensitic plates with an acicular morphology. These results are in accordance with the studies of [27,29]. The microstructure of the TIG-cladded sample consisted of α plates separated by β phase. The individual α plates were visible. Due to the large heat input and slow cooling rates, it resulted in a mixture of martensitic and diffusionally transformed microstructures. A Colony-type microstructure was predominant, as observed by [30]. These results have been provided in Figure 7. The microstructure significantly affects the tribological properties of a material. Due to high hardness and elevated strength of α’s martensitic microstructure, the EDM-ed samples exhibited more resistance to deformation under load as compared to the TIG-cladded sample, resulting in superior tribological performance.

### 3.3. Microhardness

The effect of EDM parameters on microhardness was studied. Microhardness increased with higher discharge current due to the high process temperature, which increased carbon content and triggered the formation of carbides. Similarly, microhardness increased with longer pulse on time, as higher values promoted the deposition of melted material. This melted material had a different chemical composition than the substrate material, resulting in a harder recast layer. Discharge current had a smaller impact on microhardness compared to pulse on time. The graphite electrode produced a harder recast layer than the copper electrode due to higher carbon content and carbide formation [31]. These findings were confirmed by XRD and EDX results, as shown in Figure 8a.

Taguchi optimization (Figure 8b and Table 5) revealed that pulse on time had the highest contribution (37.71%) to microhardness, followed by discharge current (35.60%) and electrode material (26.68%). The optimal microhardness was achieved using a graphite electrode at 400 µs pulse on time and 20 A discharge current. The difference between the microhardness of graphite-coated and copper-coated specimens was minimal. The possible reason is the almost same diffraction patterns exhibited by these specimens.

On the other hand, average microhardness of the TIG-cladded sample was 1052 ± 10 HV, likely due to the presence of hardening precipitates [32]. Moreover, the suitable process parameters and comparatively fine coated surface contributed to the enhanced microhardness [33]. The optimized EDM-ed sample exhibited 10.93% higher microhardness than the TIG-cladded sample, as shown in Figure 8c. These results suggest that EDM recast layers can serve as effective coatings.

### 3.4. Tribological Performance

The tribological performance was determined—in terms of coefficient of friction, specific wear rate, and worn surface morphology. The COF of all samples was determined and compared after tribological testing, as shown in Figure 9. The surface roughness affected the coefficient of friction values. The EDM-ed samples had higher surface roughness as compared to the TIG-cladded sample and the substrate material, which resulted in a sharp decrease in COF at the initial stage. The copper-coated sample initially showed a low COF due to the presence of ridges. The COF decreased up to a sliding distance of 0.34 m. After 4.72 m sliding distance, the surface became even, and the COF increased, approaching that of the substrate material. The graphite-coated sample also exhibited a decreasing COF trend at the start of testing. When the sliding distance reached 0.48 m, the COF increased, becoming almost constant at a 4.92 m sliding distance. Overall, it exhibited a lower COF compared to the substrate material. The low COF in EDM-ed samples was attributed to micro-fragmentation, which resulted in brittle detachment and reduced friction [34].

In the TIG-cladded sample, a sharp decrease in COF was not prominent at the beginning due to the lower surface roughness. The COF values were higher than those of the graphite-coated sample up to a sliding distance of 6.09 m, after which they decreased. The average COF of the TIG-cladded sample was between that of the graphite-coated and copper-coated samples. The reduction in COF was due to the higher distribution of carbides and silicides. Another reason is the increased surface hardness, which reduced load transfer to the specimen surface [35]. The optimized EDM recast layers exhibited lower COF compared to the TIG-cladded sample (Table 6), confirming their potential as effective coatings.

The specific wear rate was calculated using the cumulative wear volume, applied load, and sliding distance. The specific wear rates of the graphite-coated, TIG-cladded, and copper-coated samples were lower than that of the substrate material. Among the coated samples, the graphite-coated sample exhibited the best wear resistance, while the copper-coated sample showed the worst. The TIG-cladded sample performed intermediately, as shown in Figure 10.

The lower specific wear rates of the EDM-ed samples were attributed to the presence of carbides and oxides in the recast layers. Moreover, the recast layers exhibited high hardness and ceramic-like characteristics, resulting in lower specific wear rates [36]. The TIG-cladded sample showed improved wear resistance due to the silicon carbide powder embedded in the substrate, as well as titanium silicide formation [37]. The specific wear rate of the TIG-cladded sample was 7.64 ± 0.09 × 10^−7^ mm^3^/Nm, while the optimized EDM recast layer exhibited a specific wear rate of 7.52 ± 0.07 × 10^−7^ mm^3^/Nm (Table 6). These results demonstrate the superior tribological performance of EDM recast layers compared to TIG cladding coatings.

SEM analysis of the worn surfaces revealed different wear mechanisms. The substrate material (Figure 11a) exhibited micro-plowing, plastic deformation, and micro-grooves. Micro-plowing was found because of the chemically reactive nature and better surface finish of the specimen. Plastic deformation was possibly caused by the reduction in yield strength at elevated temperatures. The micro-groves were formed by the penetration of hard debris into the specimen surface. These wear appearances indicated the abrasion wear mechanism. The sub-mechanism was micro-plowing. Here, material loss occurred due to several abrasive particles acting simultaneously. The material plowed aside by-passed particles and broke off by low cycle fatigue. Usually, its common wear mechanism in ductile material results in the high deformation of a worn surface.

The copper-coated sample (Figure 11b) showed grooves, micro-fragmentation, and micro-cracks. The coated surface was highly deformed and removed. These findings were also supported by the considerable specific wear rate of the coated specimen. Abrasive wear and micro-cutting were found as the primary mechanisms. Here, material loss was exhibited proportionally to the volume of wear grooves. In contrast, the graphite-coated sample (Figure 11c) exhibited fewer signs of wear, with micro-cracks and light scratches. Micro-cracks occurred due to the highly concentrated stresses imposed by abrasive particles on the surface of brittle material. The frequent micro-cracks suggested abrasive wear and micro-cracking as the dominant mechanisms. Material loss was due to crack formation and propagation [38]. The TIG-cladded sample (Figure 11d) showed minute pitting, mild striations, and grooves, with abrasion, micro-cutting, and micro-plowing as the primary wear mechanisms [39].

The graphite-coated sample, with its higher carbon content, showed greater resistance to abrasion and better tribological performance compared to the TIG-cladded sample. The minimal specific wear rate and light scratches on the graphite-coated sample further supported these observations. The significant differences in worn surface morphology between the graphite-coated and TIG-cladded sample highlight the potential of EDM recast layers as effective coatings.

### 3.5. Worn Products

Energy-dispersive X-ray spectroscopy and X-ray diffraction analyses were conducted to determine different worn products. EDX revealed changes in the chemical composition of the coated samples. The copper-coated sample (Figure 12a) showed the addition of copper due to its reaction with the dielectric, as well as carbon from the decomposition of EDM oil. Oxygen was added due to the oxidation of melted material, while titanium and vanadium content decreased. The decreased constituent elements of the substrate material were increased by the electrode and dielectric [40]. The graphite-coated sample (Figure 12b) had a high carbon content (52.78% by weight), resulting from the decomposition of the graphite electrode and EDM oil. The chemical composition of the substrate material has been provided in Figure 12c.

The TIG-cladded sample (Figure 13a) showed the addition of oxygen (28.02% by weight), carbon (13.66% by weight), and silicon (2.39% by weight) in the cladding layer due to the embedded silicon carbide powder. On the other hand, titanium, aluminum, and vanadium were reduced. When reaching the interface, the diffusion of atoms altered. Carbon and oxygen were reduced while silicon content increased. The interface exhibited oxygen, silicon, and carbon with weights of 10.08%, 4.031%, and 3.07%, respectively. In contrast to the cladding layer, the constituent elements of TC4 were increased at the interface (Figure 13b).

The changes in chemical composition, particularly the formation of carbides in the graphite-coated sample, improved the tribological performance. These findings provide a basis for developing a framework for using EDM recast layers as effective coatings.

XRD analysis (Figure 14) confirmed the formation of new compounds in the EDM-ed samples. The copper-coated sample showed the presence of Ti, TiC, and Ti_8_O_15_, with Ti_8_O_15_ improving wear resistance [34]. The graphite-coated sample exhibited TiC and Ti_6_C_3.75_, formed due to the chemical reaction between the electrode material and dielectric medium. Another reason was the migration of electrode material on the recast layer. These carbides have thermal stability at elevated temperatures, high hardness, low COF, and better resistance to wear. They contributed to the superior tribological performance of the recast layer [41]. The TIG-cladded sample showed the presence of Ti_5_Si_4_, Ti_0.75_V_0.25_, Cu_0.4_V_2_O_5_, Al_11_Ti_5_, and SiC. Ti_5_Si_4_ and SiC improved wear resistance [37]. The XRD pattern of the substrate material revealed the presence of Ti, consistent with previous studies [42]. The diffraction angles and JCPD card numbers of these compounds have also been provided in Table 7.

The formation of new compounds during EDM, particularly carbides, resulted in improved microhardness and wear resistance compared to TIG cladding coatings. These findings confirm that EDM recast layers can serve as effective coatings.

### 3.6. Surface Roughness

The effect of pulse on time and discharge current on surface roughness was analyzed. An increase in either resulted in an increase in the surface roughness. The effects of pulse on time were more pronounced because it dictates discharge energy, which in turn causes migration of carbon to EDM-ed surface in the form of carbides. These carbides increase the surface roughness. Similarly, higher discharge current increases the vaporization of the material being EDM-ed, which results in deep and large craters. As a result, the surface becomes rougher [43,44]. The surface produced by graphite electrode was rougher than that by copper electrode. This could possibly be due to the formation of carbides after graphite electrode was decomposed. These results have been provided in Figure 15a.

Taguchi optimization revealed that among all parameters, pulse on time is the most influential parameter. Discharge current has the second rank, and electrode material has the third rank (Figure 15b). The contribution of these parameters is 52.17%, 34.23%, and 13.58%, respectively. The optimal surface roughness was obtained from copper electrode at 100 μs pulse on time and 10 A discharge current, as shown in Table 5. On the other hand, the average surface roughness of the TIG-cladded sample was 3.289 ± 0.03 µm. The comparison of surface roughness of substrate material, selected EDM-ed samples, and the TIG-cladded sample has been provided in Figure 15c. There was a considerable difference in the surface roughness of the substrate material, EDM-ed specimens, and the TIG-cladded specimen. The substrate material with polished surface resulted in the least surface roughness. The suitable value of current made it possible that TIG cladding coatings did not burn off along with comparatively smooth surface. The higher surface roughness of EDM-ed specimens was due to uneven surface morphology caused by intense heat. The TIG-cladded sample produced less surface roughness than the optimized EDM-ed sample (trial 18). However, its tribological performance was comparatively inferior. Different combinations of pulse on time, discharge current, and electrodes produced different surface roughness. They acted as a basis to develop a framework for the use of EDM recast layers as effective coatings.

## 4. Conclusions

In the past, conventional coatings have been used for enhancing tribological performance of the machined components. Later, a few efforts have also been made to explore and use EDM recast layers for this purpose. However, the enhancement was achieved only in comparison to the substrate material, and none of the studies have compared it with TIG cladding coatings. Moreover, many lack a simultaneous focus on electrical and non-electrical parameters during EDM. This lack of focus can often result in underestimating the applications and usage cases for recast layers. Hence, the current work addresses this significant gap for the TC4 substrate. It establishes the need for more systematic studies on the use of recast layers as effective coatings. It will be more cost-effective, more sustainable, and less time-consuming. The following research findings can be inferred from this study:The optimized recast layer was created using a graphite electrode at 20 A discharge current and 400 μs pulse on time. This layer showed an 11.11% lower COF, a 1.57% lower specific wear rate, and a 10.93% higher microhardness compared to TIG cladding coatings.The optimized recast layer was also depicted with an altered microstructure, chemical composition, and phase distribution, which resulted in superior tribological performance as compared to the TIG cladding coatings. However, surface roughness, recast layer thickness, and surface morphology of the TIG cladding coating were comparatively better.Moreover, the optimized recast layer exhibited superior microhardness (289%), higher wear resistance (74.69%), and lower COF (24.52%) than the substrate material.The key factors affecting the tribological performance of the recast layers were pulse on time (electrical) and the graphite electrode (non-electrical).

This study shows that EDM recast layers are a promising alternative to conventional TIG cladding coatings. They offer better tribological performance, as well as potential cost and time savings. Future research should explore the applicability of these findings to other materials and industrial applications, as well as investigate the corrosion resistance of EDM recast layers.

## Figures and Tables

**Figure 1 micromachines-16-00913-f001:**
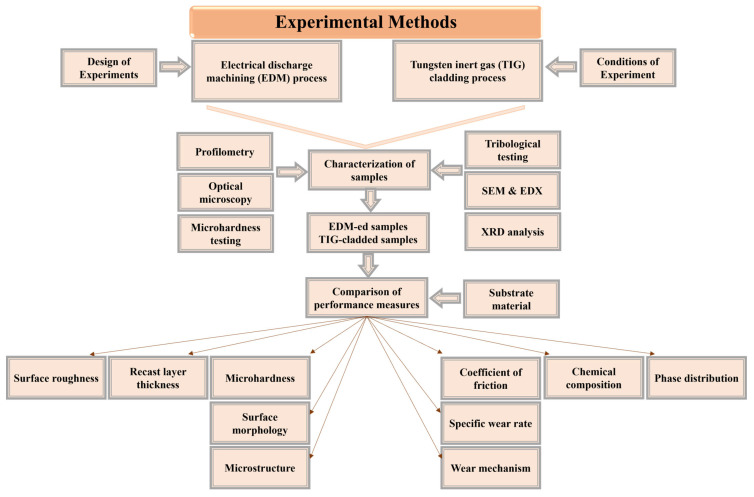
Flow diagram of experimental methodology.

**Figure 2 micromachines-16-00913-f002:**
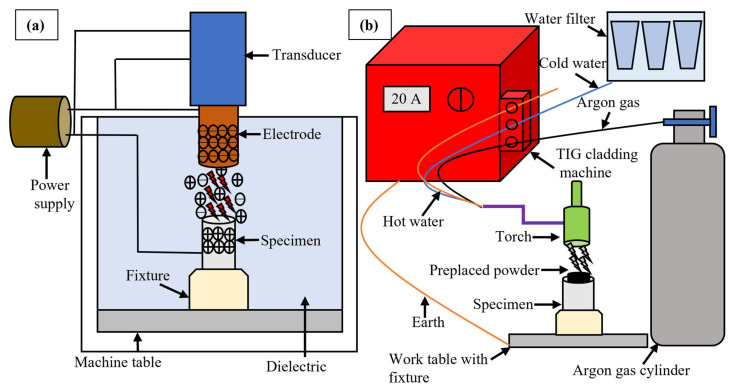
Schematic arrangement of (**a**) EDM and (**b**) TIG cladding coating.

**Figure 3 micromachines-16-00913-f003:**
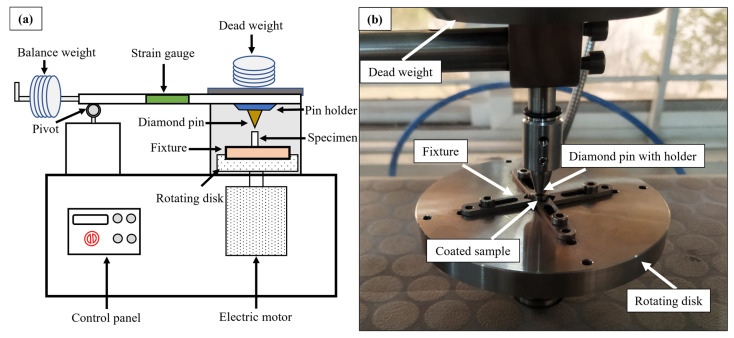
(**a**) Schematic arrangement and (**b**) experimental setup of a pin-on-disk tester.

**Figure 4 micromachines-16-00913-f004:**
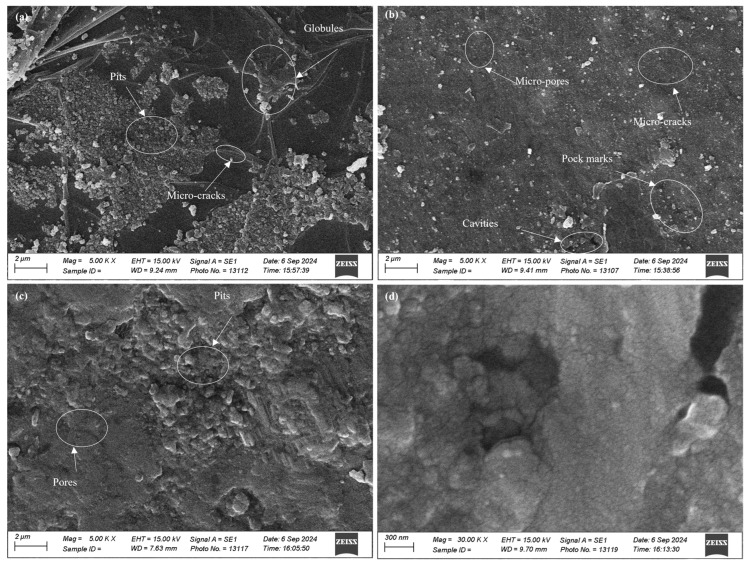
SEM micrographs of (**a**) a copper-coated sample, (**b**) a graphite-coated sample, (**c**) a TIG-cladded sample, and (**d**) the substrate material.

**Figure 5 micromachines-16-00913-f005:**
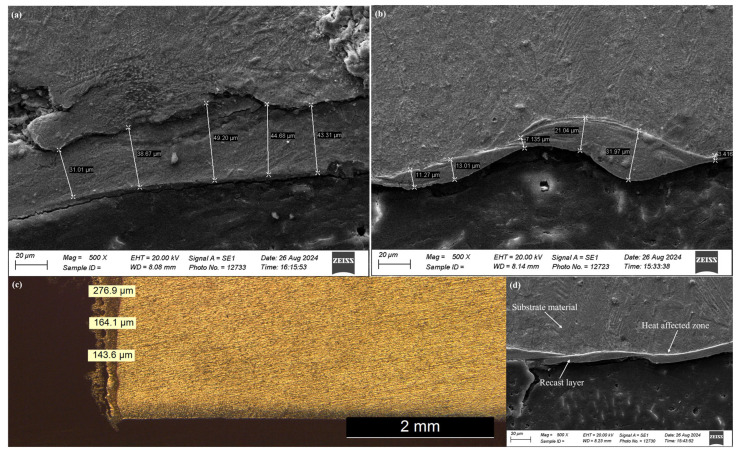
Recast layer thickness of (**a**) a graphite-coated sample, (**b**) a copper-coated sample, and (**c**) a TIG-cladded sample, as well as a (**d**) cross-sectional investigation of a EDM-ed sample.

**Figure 6 micromachines-16-00913-f006:**
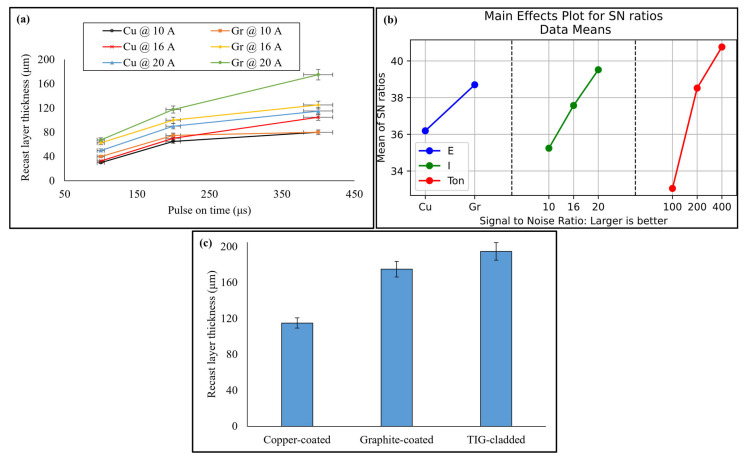
(**a**) Recast layer thickness with the variation in EDM parameters and (**b**) parameter optimization, as well as a (**c**) comparison of recast layer thickness.

**Figure 7 micromachines-16-00913-f007:**
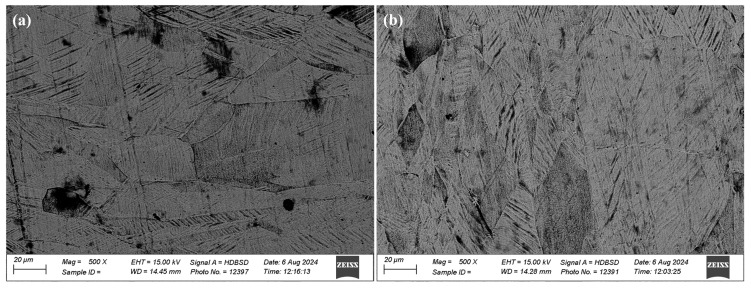
Microstructure of an (**a**) EDM-ed sample (graphite-coated) and a (**b**) TIG-cladded sample.

**Figure 8 micromachines-16-00913-f008:**
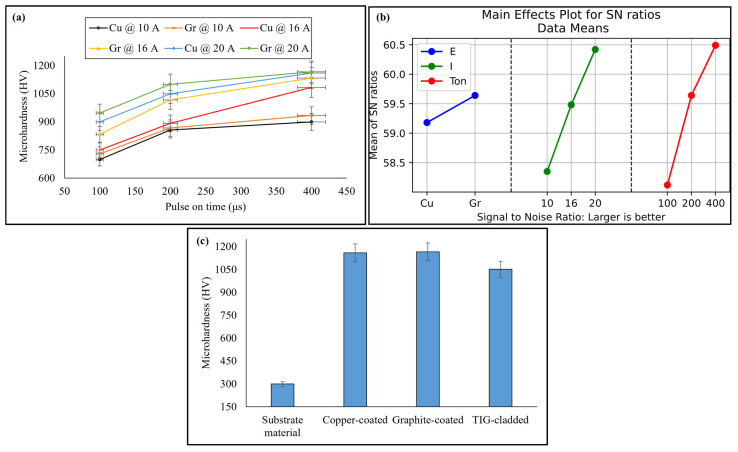
(**a**) Microhardness with the variation in EDM parameters and (**b**) parameter optimization, as well as a (**c**) comparison of microhardness.

**Figure 9 micromachines-16-00913-f009:**
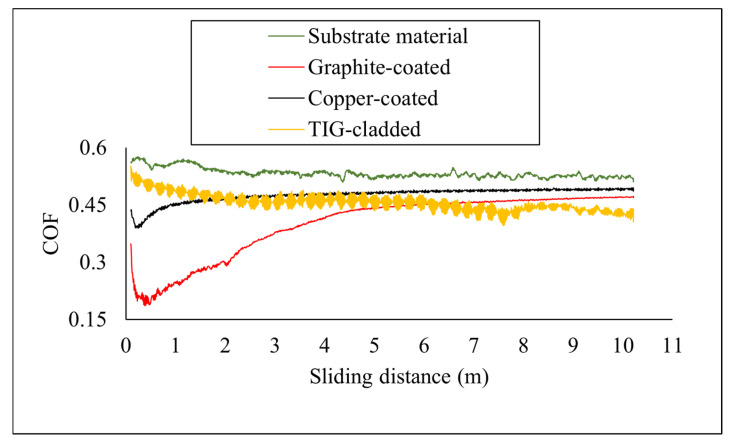
Coefficient of friction of the coated samples and substrate material.

**Figure 10 micromachines-16-00913-f010:**
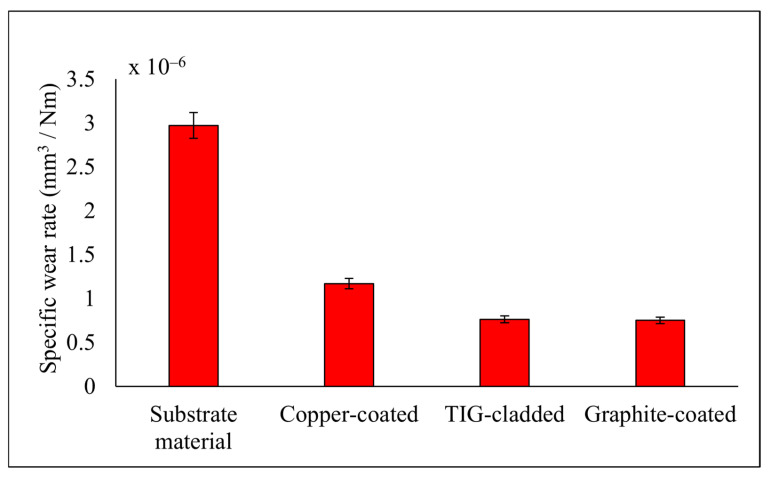
Specific wear rate of the coated samples and substrate material.

**Figure 11 micromachines-16-00913-f011:**
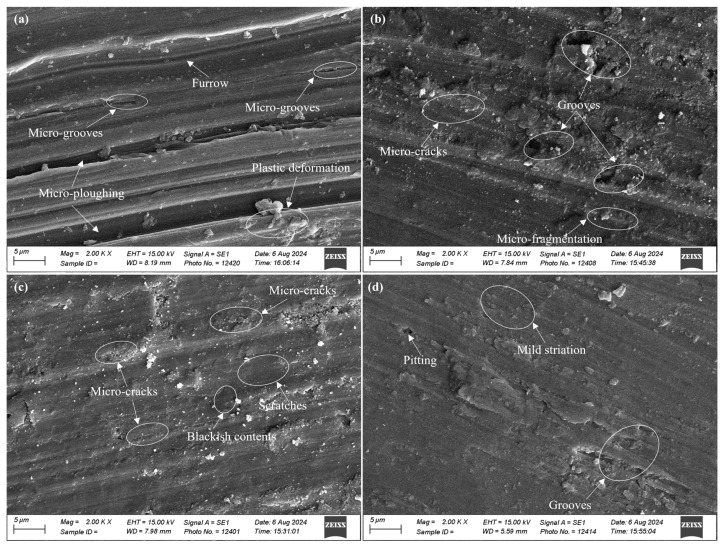
SEM micrographs after the tribological testing of (**a**) copper-coated sample, (**b**) a graphite-coated sample, (**c**) a TIG-cladded sample, and (**d**) the substrate material.

**Figure 12 micromachines-16-00913-f012:**
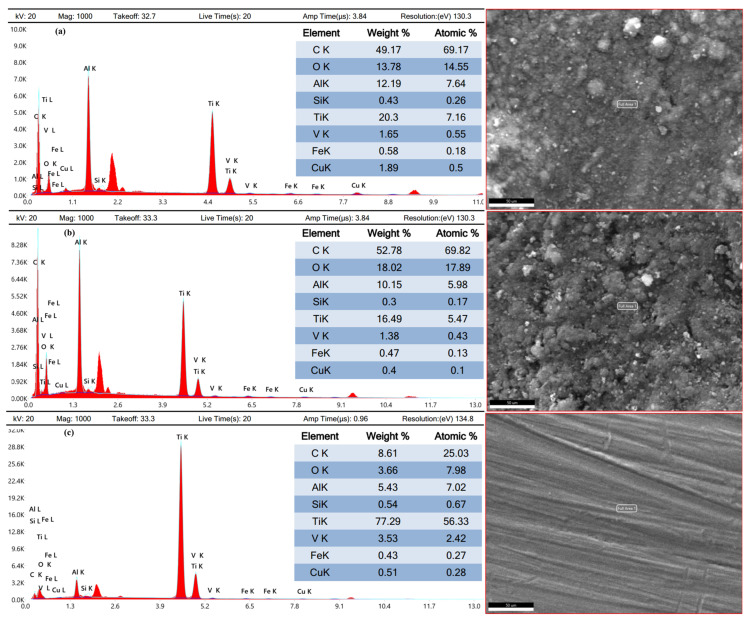
Energy-dispersive X-ray analysis of (**a**) a copper-coated sample, (**b**) a graphite-coated sample, and the (**c**) substrate material.

**Figure 13 micromachines-16-00913-f013:**
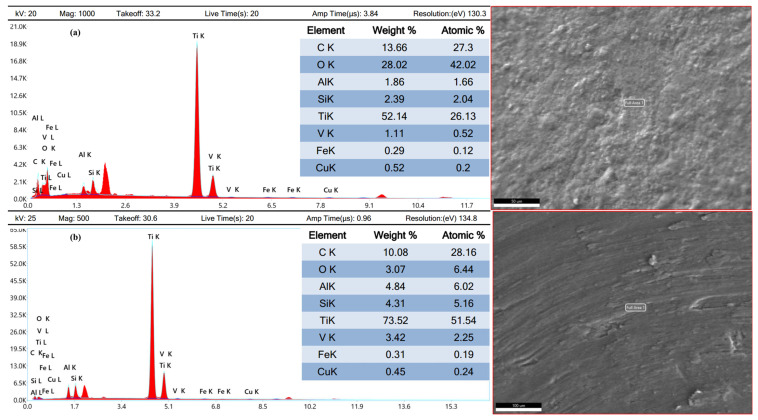
Energy-dispersive X-ray analysis of (**a**) a cladding layer and (**b**) the interface of a TIG-cladded sample.

**Figure 14 micromachines-16-00913-f014:**
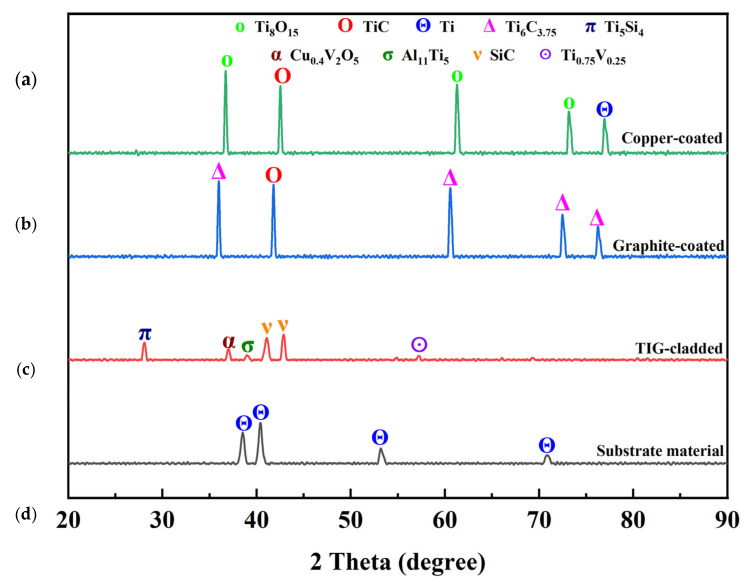
X-ray diffraction patterns of (**a**) a copper-coated sample, (**b**) a graphite-coated sample, (**c**) a TIG-cladded sample, and (**d**) the substrate material.

**Figure 15 micromachines-16-00913-f015:**
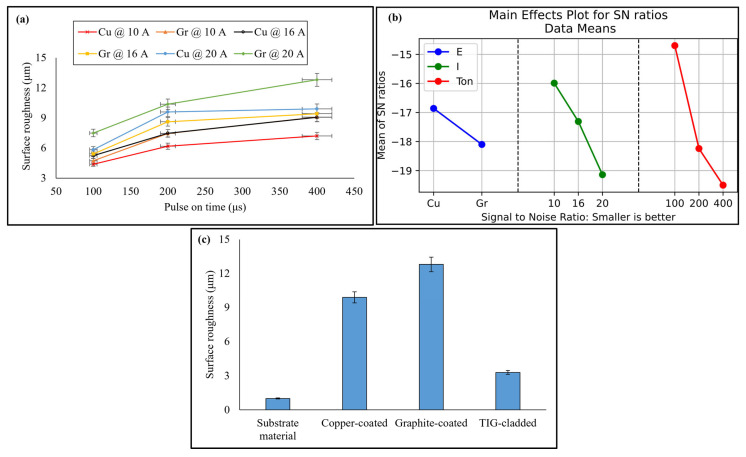
(**a**) Surface roughness with the variation in EDM parameters and (**b**) parameter optimization, as well as (**c**) comparison of surface roughness.

**Table 1 micromachines-16-00913-t001:** Specifications of materials used for experimentation.

Nomenclature	Material	Specifications
Specimen	TC4 (5.13 wt. % Al, 0.256 wt. % Si, 90.94 wt. % Ti and 4.07 wt. % V)	Diameter = 8 mm, H_m_ = 300 HV and R_a_ = 1 μm
EDM electrodes	Copper (0.455 wt. % Si, 0.074 wt. % Fe, 96.66 wt. % Cu, 2.79 wt. % Zn and 0.025 wt. % Pb)	Diameter = 10 mm, H_m_ = 100 HV and R_a_ = 0.79 μm
	Graphite (98.97 wt. % C, 0.236 wt. % Al, 0.354 wt. % Si, 0.095 wt. % S and 0.343 wt. % Fe)	Diameter = 10 mm, H_m_ = 21 HV and R_a_ = 1.25 μm
TIG cladding electrode	Tungsten thoriated (0.207 wt. % Ni, 0.801 wt. % Zn, 0.222 wt. % Zr, 0.078 wt. % Ni, 95.79 wt. % W and 2.90 wt. % Bi)	Diameter = 2.4 mm
Counter-body	Diamond (0.041 wt. % B, 99.93 wt. % C and 0.026 wt. % N)	Diameter = 3 mm

**Table 2 micromachines-16-00913-t002:** Experimental parameters for EDM.

Parameters for DOE	Levels		
	Level 1	Level 2	Level 3
Discharge current (A)	10	16	20
Pulse on time (μs)	100	200	400
Electrodes	Copper	Graphite	-
**Other Parameters**	**Description**		
Pulse off time (μs)	100		
Gap (mm)	1		
Duty factor (%)	75		
Dielectric	406 EDM oil		
Polarity	Positive		

**Table 3 micromachines-16-00913-t003:** Input variables for TIG cladding coating and their values.

Input Parameters	Current (A)	Voltage (V)	Powder Size (μm)	Speed (mm/s)
Values	20	15	110	1

**Table 4 micromachines-16-00913-t004:** Experimental conditions for tribological testing.

Input Parameters	Values
Applied load (N)	50
Track radius (mm)	3
Rotational speed (RPM)	100
Sliding distance (m)	10
Temperature (°C)	25
Humidity (%)	RH 47

**Table 5 micromachines-16-00913-t005:** Response table for signal-to-noise ratio for Taguchi optimization.

Level	h_c_ (Larger Is Better)			H_m_ (Larger Is Better)			R_a_ (Smaller Is Better)		
	E	I	T_ON_	E	I	T_ON_	E	I	T_ON_
1	36.19	35.24	33.05	59.18	58.35	58.12	−16.86	−15.99	−14.70
2	38.70	37.57	38.52	59.64	59.48	59.64	−18.10	−17.31	−18.24
3		39.52	40.76		60.42	60.49		−19.14	−19.50
Delta	2.51	4.28	7.71	0.46	2.07	2.37	1.25	3.15	4.80
Rank	3	2	1	3	2	1	3	2	1

**Table 6 micromachines-16-00913-t006:** Comparison of tribological performance of the EDM-ed samples, TIG-cladded sample, and substrate material.

Sample Nomenclature	COF	Wear Volume (mm^3^)	Specific Wear Rate (mm^3^/Nm)
Substrate material	0.53 ± 0.03	0.001486	(2.972 ± 0.28) × 10^−6^
Copper-coated	0.47 ± 0.01	0.000586	(1.172 ± 0.08) × 10^−6^
TIG-cladded	0.45 ± 0.04	0.000382	(7.64 ± 0.09) × 10^−7^
Graphite-coated	0.40 ± 0.02	0.000376	(7.52 ± 0.07) × 10^−7^

**Table 7 micromachines-16-00913-t007:** XRD analysis of the EDM-ed samples, TIG-cladded sample, and substrate material.

Symbol	Chemical Formula	JCPD Card	Diffraction Angle (2ϴ°)
Δ	Ti_6_C_3.75_	79-0971	35.936, 60.457, 72.378, 76.190
O	TiC	32-1383	41.710, 41.710
Π	Ti_5_Si_4_	23-1079	28.036
A	Cu_0.4_V_2_O_5_	18-0464	37.200
Σ	Al_11_Ti_5_	42-1135	39.049
V	SiC	12-1301	41.186, 42.823
ʘ	Ti_0.75_V_0.25_	51-0636	57.244
Θ	Ti	44-1294	38.765, 40.170, 53.004, 70.660, 76.218,
O	Ti_8_O_15_	50-0790	36.120, 60.870, 72.491

## Data Availability

The data used to support the findings of this study are included within the article.

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
