# Peer review of "Tribology of EDM Recast Layers Vis-À-Vis TIG Cladding Coatings: An Experimental Investigation"

_micromachines, 2025, doi:10.3390/mi16080913_

Round 1
Reviewer 1 Report
Comments and Suggestions for Authors
The article is well prepared, only some of the minor issue, such as do authors calculate wear volume ?
Whats the coating thickness on the substrate?
Do author correlated coating thickness, wear volume and hardness ?
Author Response
Thank you very much for taking the time to review this manuscript. Please find the detailed responses below and the corresponding revisions highlighted in the re-submitted files.
|
Comments 1: [The article is well prepared, only some of the minor issue, such as do authors calculate wear volume ?] |
Response 1: [Wear volume was calculated and mentioned in Table 6, page 15.]
|
Comments 2: [Whats the coating thickness on the substrate?] |
Response 2: [The substrate material has no coating thickness. The coating was produced after the EDM process. It has been mentioned in section 2.2.1, lines 116~119, page number 4.]
|
Comments 3: [Do author correlated coating thickness, wear volume and hardness ? ] |
Response 3: [No direct correlation was found between coating thickness, wear volume, and hardness of EDM-ed coating and TIG-cladding coating.]
|
Reviewer 2 Report
Comments and Suggestions for Authors
- EDM and TIG should give full name(tungsten inert gas, electrical discharge machining) in the abstract section.
- Pls give reasons to explain why select Copper, Graphite, as research objects.
- Pls give chemical composition of TC4, Copper, Graphite, a tungsten thoriated electrode Diamond.
- Pls give reasons to explain why select test parameters as Table 4.
- In Figure.4 it should be “substrate”but not “substarte”
- Scale should be added in the Fig.5(c).
- The surface morphologies of copper- coated materials should also expressed same as Fig.5.
- The EDS mapping and lining scan should be conduct to study the atom diffusion of cladding layer and interface.
- Section 3.4 &3.5 should introduce firstly in section 3 results and discussion.
- In Fig.9, which sample did the authors express? copper- coated of graphite- coated?
- The analysis of COF should be studied combine with wear rate, worn surface, and even wear debris.
- The section 3.9 & 3.10 should be combined for a section named worn products.
- The involved worn mechanism should be deeply discussed.
Author Response
Thank you very much for taking the time to review this manuscript. Please find the detailed responses below and the corresponding revisions highlighted in the re-submitted files.
|
||
Comments 1: [EDM and TIG should give full name(tungsten inert gas, electrical discharge machining) in the abstract section.] |
||
Response 1: [All abbreviations removed from lines 15~16,18,22~23,27~29 of abstract section.]
|
||
Comments 2: [Pls give reasons to explain why select Copper, Graphite, as research objects.] |
||
Response 2: [It is evident from literature that copper and graphite electrodes are frequently used in mold industry [1], [2]. So, they were employed for EDM in the present study. It has been mentioned in section 2.2.1, lines 124~125, page 4.]
|
||
Comments 3: [Pls give chemical composition of TC4, Copper, Graphite, a tungsten thoriated electrode Diamond.] |
||
Response 3: [The required chemical composition has been added in Table 1, page 3.]
|
||
Comments 4: [Pls give reasons to explain why select test parameters as Table 4.] |
||
Response 4: [It has been mentioned in section 2.2.3, lines 189~196, page 6 [“Usually, tribological tests are carried out for a specific application. EDM and TIG cladding have versatile applications such as automotive, biomedical and aerospace industries. In this regard, many combinations of counter body material, applied load, track radius, rotational speed, and sliding distance were possible. As the indentation exhibits the wear response of the recast layers rather than specific application. Several experiments were performed to choose the experimental parameters. It was intended to obtain a measurable and suitable wear section from the heat affected zone. The experimental conditions based on pilot trials have been provided in Table 4”.] |
Comments 5: [In Figure.4 it should be “substrate”but not “substarte”] |
Response 5: [The typos removed from said figure, renumbered as Figure 15 and placed on page 21.]
|
Comments 6: [Scale should be added in the Fig.5(c).] |
Response 6: [Figure 5c representing cross-sectional investigation of EDM-ed sample has been replaced with Figure 5d along with scale on page 10.] |
Comments 7: [The surface morphologies of copper- coated materials should also expressed same as Fig.5.] |
Response 7: [Figure 5b added, representative of morphologies of copper-coated specimen on page 10.]
|
Comments 8: [The EDS mapping and lining scan should be conduct to study the atom diffusion of cladding layer and interface.] |
Response 8: [A comprehensive EDX analysis at three different locations of cladding layer and interface was conducted. The presence of different atoms at these locations was thoroughly studied. The results have been mentioned in section 3.5, lines 390~397, pages 15~16 and Figure 13, page 18.] |
Comments 9: [Section 3.4 &3.5 should introduce firstly in section 3 results and discussion.] |
Response 9: Thank you for pointing this out. We agree with this comment. Therefore, [section 3.4 and section 3.5 have been placed at the start of section 3.]
|
Comments 10: [In Fig.9, which sample did the authors express? copper- coated of graphite- coated?] |
Response 10: [Figure 9 has been renumbered as Figure 7 and it expresses graphite-coated specimen. It has also been mentioned in the caption on page 11.] |
Comments 11: [The analysis of COF should be studied combine with wear rate, worn surface, and even wear debris.] |
Response 11: Thank you for pointing this out. We agree with this comment. Therefore, [the sections coefficient of friction, specific wear rate and worn surface morphology combined and named tribological performance.] |
Comments 12: [The section 3.9 & 3.10 should be combined for a section named worn products.] |
Response 12: Thank you for pointing this out. We agree with this comment. Therefore, [section 3.9 and section 3.10 combined and named as worn products.]
|
Comments 13: [The involved worn mechanism should be deeply discussed.] |
Response 13: [The involved wear mechanisms further elaborated in section 3.4, lines 357~359, 363~364, 368~369, pages 13~14.] |
References
[1] B. Ekmekci and E. Güngör, “A comparative study on the wear resistance of electrical discharge machined surfaces,” Machining Science and Technology, vol. 21, no. 4, pp. 632–650, Oct. 2017, doi: 10.1080/10910344.2017.1336180.
[2] M. Adnan, W. Qureshi, M. Umer, and D. Botto, “Tribological Characterization of Electrical Discharge Machined Surfaces for AISI 304L,” Materials, vol. 15, no. 3, p. 1028, Jan. 2022, doi: 10.3390/ma15031028.
Reviewer 3 Report
Comments and Suggestions for Authors
The subject of the manuscript is interesting, but there are a number of questions and comments on the contents of the manuscript:
1) In the "Abstract" section, it is necessary to provide an explanation of the abbreviations.
2) Why was it necessary to compare the coatings obtained by these two different methods? Is one of the methods more effective and less expensive?
3) Section 2.2 and Section 2.2.3 duplicate each other. They need to be combined.
4) Fig. 4 (c). What is the reason for such different roughness? The discussion should be added to the text of the manuscript.
5) Fig. 5 is of poor quality. It is necessary to redo the sections, possibly etch them, and then provide the results of the layer thickness study. Perhaps, sections 3.2, 3.5 and 3.9 should be combined. These sections, as well as Section 3.10, should be presented before the roughness study. Because the obtained coating is demonstrated first, and only then its features are studied.
6) Fig. 7 . The microhardness of the surface layer obtained in different ways is slightly different. What is the reason? The discussion should be added to the text of the manuscript.
7) Section 3.6. It turns out that roughness directly affects the friction coefficient, especially at the initial stage. The discussion should be added to the text of the manuscript.
8) Fig. 13 is very imperceptible to the eye, small font. Maybe it would be better to present the chemical composition of the surface in the form of a table?
9) The scientific novelty of the study is not very clear.
Author Response
Thank you very much for taking the time to review this manuscript. Please find the detailed responses below and the corresponding revisions highlighted in the re-submitted files.
|
Comments 1: [In the "Abstract" section, it is necessary to provide an explanation of the abbreviations.] |
Response 1: [All abbreviations explained on lines 15, 16, 18, 22, 23, 27~29 in the abstract section.]
|
Comments 2: [Why was it necessary to compare the coatings obtained by these two different methods? Is one of the methods more effective and less expensive?] |
Response 2: [The coatings obtained by two different methods were compared to explore the potential of EDM recast layers to reduce the costs and time required for post-machining TIG-cladding coatings. It has been mentioned in section 1, lines 64~66, page 2.]
|
Comments 3: [Section 2.2 and Section 2.2.3 duplicate each other. They need to be combined.] |
Response 3: Thank you for pointing this out. We agree with this comment. Therefore, [the duplicated text removed from section 2.2.]
|
Comments 4: [Fig. 4 (c). What is the reason for such different roughness? The discussion should be added to the text of the manuscript.] |
Response 4: [The Figure has been renumbered as Figure 15c. The reason for different surface roughness of the coated samples and the substrate material has been mentioned in section 3.6, lines 444~449, page 20.] |
Comments 5: [Fig. 5 is of poor quality. It is necessary to redo the sections, possibly etch them, and then provide the results of the layer thickness study. Perhaps, sections 3.2, 3.5 and 3.9 should be combined. These sections, as well as Section 3.10, should be presented before the roughness study. Because the obtained coating is demonstrated first, and only then its features are studied.] |
Response 5: [The quality of Figure 5 enhanced. Also, section 3.2 and section 3.5 combined. Moreover, these sections have been placed before the section surface roughness.]
|
Comments 6: [Fig. 7 . The microhardness of the surface layer obtained in different ways is slightly different. What is the reason? The discussion should be added to the text of the manuscript.] |
Response 6: [The reason for the minimal difference between microhardness of copper-coated and graphite-coated samples has been added to the text in section 3.3, lines 302~304, page 12.] |
Comments 7: [Section 3.6. It turns out that roughness directly affects the friction coefficient, especially at the initial stage. The discussion should be added to the text of the manuscript.] |
Response 7: [The required discussion has been added in section tribological performance, lines 314~327, pages 12~13.]
|
Comments 8: [Fig. 13 is very imperceptible to the eye, small font. Maybe it would be better to present the chemical composition of the surface in the form of a table?] |
Response 8: Thank you for pointing this out. We agree with this comment. Therefore, [the Figure 13 has been renumbered as Figure 12 and its quality has been enhanced on page 17.] |
Comments 9: [The scientific novelty of the study is not very clear.] |
Response 9: [The scientific novelty has been mentioned in the conclusion section, lines 460~469, page 20~21.]
|
Round 2
Reviewer 2 Report
Comments and Suggestions for Authors
Accept in present form.
Author Response
Thank you very much for taking the time to review the revised manuscript.
|
Open Review: [I would not like to sign my review report.]
Response: [The Open Review option, where the review reports and authors’ responses are published alongside the paper may be changed.] |
Reviewer 3 Report
Comments and Suggestions for Authors
Before publication, authors need to finalize the manuscript design.
1) In Table 1, it is better to present the chemical composition in a more classical way. In the "Material" column, present the composition in brackets after the name of the material, and present the chemical elements as symbols to save space. For example, 5.13 wt. % Al.
2) Figs. 12 and 13. It is better to provide the composition in the empty field where the spectrum is shown, and leave the SEM photo without inscriptions so that the surface morphology can be examined.
Author Response
Thank you very much for taking the time to review the revised manuscript. Please find the detailed responses below and the corresponding revisions highlighted in the re-submitted files.
|
Comments 1: [Before publication, authors need to finalize the manuscript design. |
Response 1: Thank you for pointing this out. We agree with this comment. Therefore, [the chemical composition has been presented in “Material” column using symbols on page 3]
|
Comments 2: [Figs. 12 and 13. It is better to provide the composition in the empty field where the spectrum is shown, and leave the SEM photo without inscriptions so that the surface morphology can be examined.] |
Response 2: Thank you for pointing this out. We agree with this comment. Therefore, [in Figure 12 and 13, the chemical composition has been given in the empty field to clearly show the surface morphology on page 17 and 18, respectively.]
|